# Cost analysis of chronic pain due to musculoskeletal disorders in Chile

**Manuel A. Espinoza**[1,2☯¤]*, **Norberto Bilbeny**[3‡], **Tomas Abbott**[2☯¤], **Cesar Carcamo**[3‡], **Pedro Zitko**[1,4‡], **Paula Zamorano**[2,5¤‡], **Carlos Balmaceda**[1,2☯¤]

**1** Department of Public health, Pontificia Universidad Católica de Chile, Santiago, Chile, **2** Health Technology Assessment Unit, Center of Clinical Research, Pontificia Universidad Católica de Chile, Santiago, Chile, **3** Asociación Chilena del Dolor, Santiago, Chile, **4** Health Services and Population Research Department, IoPPN. King's College London, London, United Kingdom, **5** Centro de Innovación en Salud ANCORA UC, Facultad de Medicina, Pontificia Universidad Católica de Chile, Santiago, Chile

☯ These authors contributed equally to this work.
¤ Current address: Diagonal Paraguay, Santiago, Chile
‡ NB, CC, PZ, and PZ also contributed equally to this work.
* manuel.espinoza@uc.cl

**Data Availability Statement:** All relevant data are within the paper.

## Abstract

The magnitude of the cost of chronic pain has been a matter of concern in many countries worldwide. The high prevalence, the cost it implies for the health system, productivity, and absenteeism need to be addressed urgently. Studies have begun describing this problem in Chile, but there is still a debt in highlighting its importance and urgency on contributing to chronic pain financial coverage. This study objective is to estimate the expected cost of chronic pain and its related musculoskeletal diseases in the Chilean adult population. We conducted a mathematical decision model exercise, Markov Model, to estimate costs and consequences. Patients were classified into severe, moderate, and mild pain groups, restricted to five diseases: knee osteoarthritis, hip osteoarthritis, lower back pain, shoulder pain, and fibromyalgia. Data analysis considered a set of transition probabilities to estimate the total cost, sick leave payment, and productivity losses. Results show that the total annual cost for chronic pain in Chile is USD 943,413,490, corresponding an 80% to the five diseases studied. The highest costs are related to therapeutic management, followed by productivity losses and sick leave days. Low back pain and fibromyalgia are both the costlier chronic pain-related musculoskeletal diseases. We can conclude that the magnitude of the cost in our country's approach to chronic pain is related to increased productivity losses and sick leave payments. Incorporating actions to ensure access and financial coverage and new care strategies that reorganize care delivery to more integrated and comprehensive care could potentially impact costs in both patients and the health system. Finally, the impact of the COVID-19 pandemic will probably deepen even more this problem.

## Introduction

Multimorbidity is currently challenging health systems response. The impact it causes in terms of costs and people is driving a reorganization of care delivery [1]. In this sense, chronic pain

**Funding:** the funder for the stduy was Asociacion Chilena para el estudio del dolor y cuidados paliativos.

**Competing interests:** no

due to musculoskeletal diseases (MSKD) is highly relevant because of its elevated prevalence [2, 3]. Reports have shown that chronic pain related to MSKD affects around the 20% of the worldwide population [4]. Furthermore, MSKD is one of the leading causes of disease burden, accounting for 139 million disability-adjusted life years (DALYs) [5]. The COVID-19 pandemic has worsened these people's pain, expecting that this problem's magnitude will continue to grow even more than the pre-pandemic indicators [6, 7].

Chronic pain is defined as unceasing pain during at least three months, or pain that persists more than the average healing time, which usually has no protective function, impairs health, and generates disability [8]. There is a wide acknowledgment that pain is not the only symptom of the disease but one of the dimensions that explain the disability associated with MSKD. Moreover, various conditions and syndromes include lower back pain, osteoarthritis of the knee and hip, chronic shoulder pain, and fibromyalgia. They are associated with variable degrees of disability, determined mainly by impairments in mobility, mental health (anxiety and depression), and pain [4]. As MSKD has various treatments depending on personal needs and disability, it demands multidisciplinary care with significant utilization of health services that last in the long term [9, 10].

The current approach of the Chilean health system for people with chronic pain related to MSKD is fragmented between the primary, secondary and tertiary levels. Although there are few Chronic Pain Units in hospitals throughout the country that provide more comprehensive care, and the scale-up of a multimorbidity approach has recently begun [11], the lack of an integrated response to this health problem still persists [3]. Thus, patients constantly seek to answer their needs generating increased costs related to health services utilization [12, 13]. This scenario makes chronic pain invisible and lacks access to more effective care.

Moreover, chronic pain is a highly cost disease for the health system. Studies have shown that the US $834 million could be spent annually [2]. For example, in Chile, the estimated cost of the GDP is 0.32%, occupying about 5% of total health costs [14]. At the same time, labor sick leave and societal productivity losses add even more cost to both users and the system, being generally long-term. Unfortunately, as the health system response is fragmented, some costs may be hidden and borne by the primary care services at the expense of other activities. Thus, considering chronic pain as a disease, including in the recent CIE-11 [15], will probably allow approaching health system performance.

Then, contributing to chronic pain financial coverage is a health priority. It's urgent to make the health system's costs visible to decision-makers through a cost analysis that would potentially add value by identifying the associated cost for each dimension involved in chronic care providing a complete diagnosis of the resources involved. This study aims to estimate the expected cost of chronic pain and its related MSKD conditions in the Chilean adult population. We used the most recent nationally representative epidemiological data, updated healthcare costs, and resource use data to provide a robust estimate to inform the magnitude of this health problem.

## Materials and methods

We conducted a cost analysis study based on a mathematical decision model to estimate the cost and consequences of MSKD and pain. The mathematical model has been explained in detail elsewhere [16], but briefly, it is a cohort Markov model of four states: mild chronic pain, moderate chronic pain, severe chronic pain, and death due to other causes (Fig 1). Each health state was defined following the severity revealed by the Visual Analogue Scale (VAS) [17]. Mild pain was classified as a VAS 1–3, moderate pain as a VAS 4–6, and severe pain as a VAS 7–10 [16].

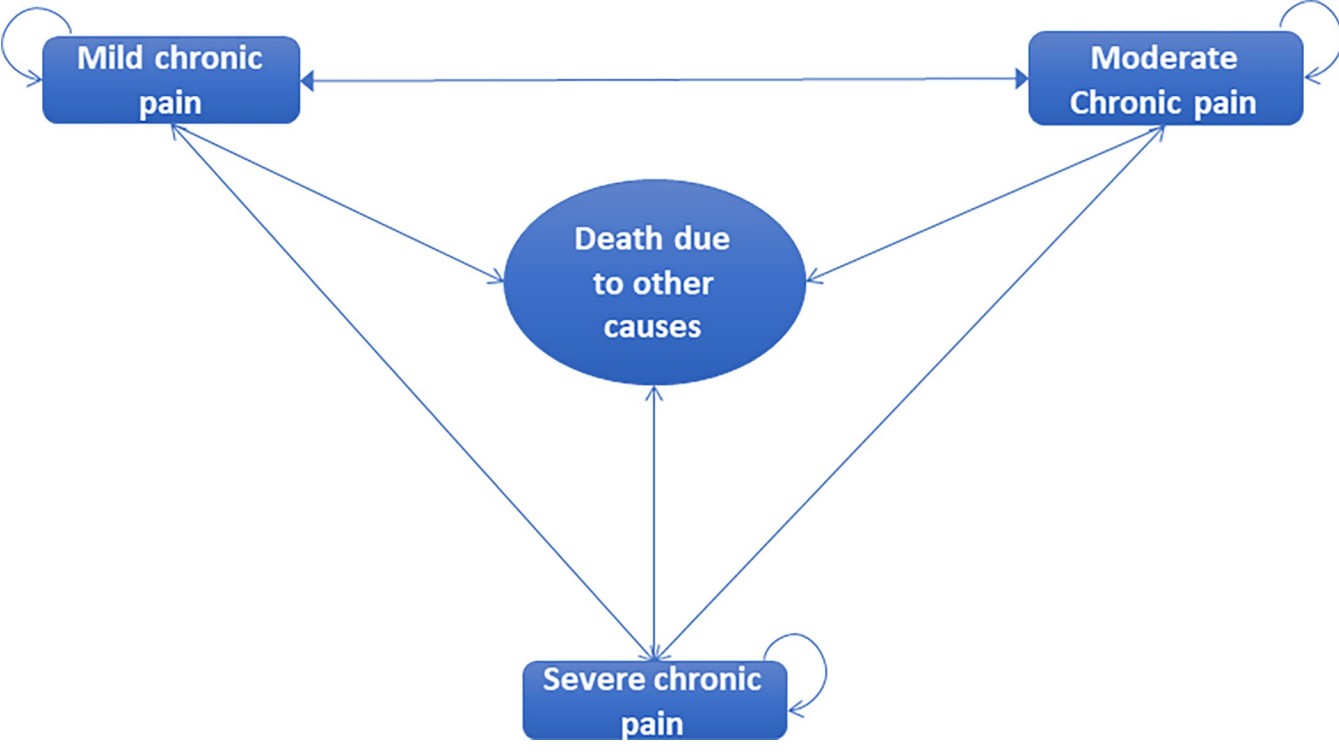

**Fig 1. Cohort Markov model structure.**

The population with MSKD was classified in clusters as mentioned above. According to an expert elicitation, we allocated 7.5% of patients to mild pain, 32.5% to moderate, and 60% to severe pain [16]. This allocation aimed to represent the observed distribution at the first consultation. The population was restricted to five diseases: knee osteoarthritis (knee OA), hip osteoarthritis (hip OA); lower back pain (LBP); shoulder pain; and fibromyalgia (FM). Although FM can be considered a central and peripheral nervous system disease, we decided to include it as an MSKD because it is a significant cause of pain referred to as the musculoskeletal system. Furthermore, we included a category of chronic musculoskeletal pain by all causes.

We modeled a hypothetical estimated cohort through the prevalence of each MSKD disorder, previously defined and estimated according to Zitko et al. [18, 19]. The data was obtained from the National Health Survey 2016–2017 (ENS 2016–7) [20]. We used a 1-month cycle validated by local experts, assuming that is a reasonable period to observe the clinical response to current treatments. The time horizon of the study was one year as the Chilean health systems budgets are annually structured; hence, no discount rates were applied.

## Data analysis

Data analysis estimated cost, sick leave, and productivity losses as shown in Fig 2. We used databases from public and private health insurance. We populated the model with the same set of transition probabilities used by Vargas et al. [16] and probability of death [21], obtained through an expert elicitation exercise with local Chilean experts. These probabilities were assumed to be the same regardless of the etiology of chronic pain. Furthermore, patients were supposed not to die due to chronic pain but only for general causes. Therefore, health

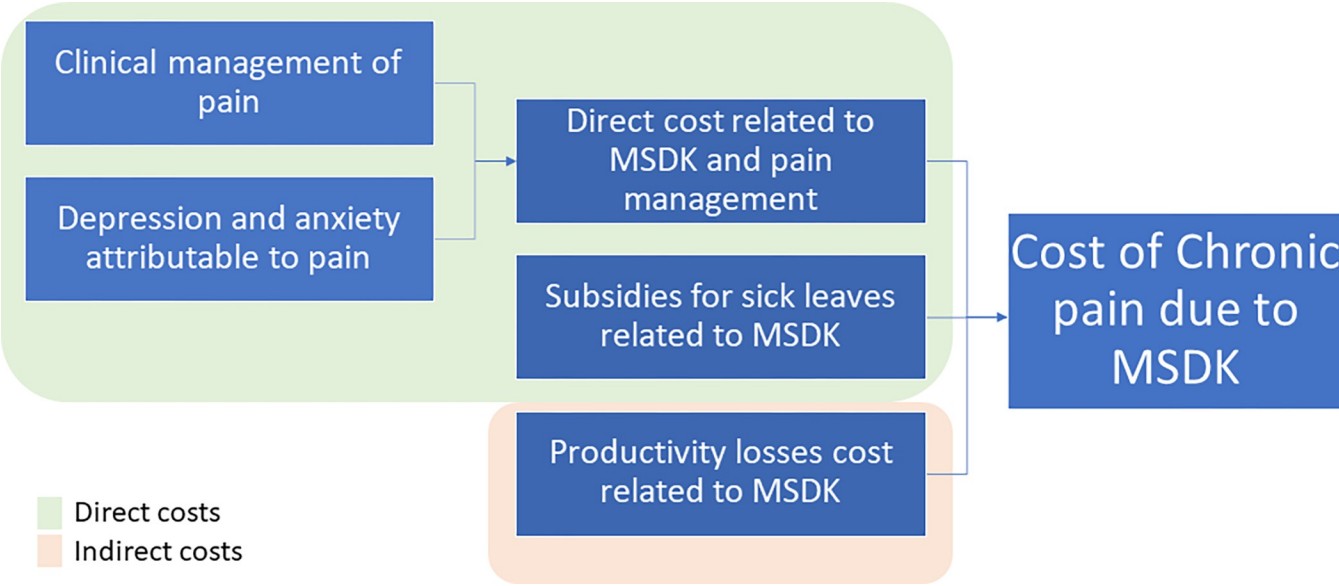

**Fig 2. Dimensions of Costs related to MSKD and chronic pain.**

impairments attributable to MSKD and pain were determined only by the health-related quality of life losses. These assumptions were considered reasonable and validated by local experts.

a. Cost Estimation: Costs related to MSKD and their attributable pain were estimated for all MSKD. Then we subtracted the sum of the cost of the five MSKD included in this study for the following domains:

- Direct costs related to the clinical management of pain included: physician appointments, pharmacological treatment ─ without considering the adverse events─ and invasive management of pain (local anesthetic infiltration).

- Depression was attributable to pain.

- Anxiety was attributable to pain.

- Subsidies for sick leave.

- Productivity losses were attributable to each group of MSKD.
  We built health baskets to treat mild, moderate, and severe chronic pain with local experts. The items identified in each basket were physician visits, physiotherapy sessions, pharmacological treatment, and hospitalizations. Then, we measured the expected frequency and quantity of utilization per month, using international and local clinical guidelines, which clinicians locally validated. Specifically for pharmaceuticals used by outpatients, we used the data of a representative telephone survey performed in the Chilean population [22]. For valuation, we used local tariffs such as the healthcare costing study of the Ministry of Health [23], the 2018 tariff of the National Fund for Health (FONASA) [24] -the Chilean public social insurer- and the prices of the central procurement processes published by the National Centre for Procurement in the case of pharmaceuticals 2018 [25].

b. Sick leave payments: We estimated the expected cost attributable to the subsidy for the number of sick leave days for each group of MSKD for public and private health insurance. It's important to mention that health insurers in Chile (public or private) pay from the 4th

day of sick leave and onwards; this means that payments are made from the 4th to the 10th day. If the sick leave is 11 days or more, then the insurance must make payments to the individual for the entire period (paying from the first sick leave day). Thus, this cost estimation includes the discount of the first three days of sick leave. We worked with two private and public databases. For the private, we did a direct estimation from the database. The public database does not report the sick leave payment. We made an additional estimation of the product between the sick leave days payments and the average daily wage of an individual affiliated on each of the four categories defined by FONASA (i.e., A, B, C, and D, being A the category with most poorer people and D the less). Then we multiplied each of these values by the expected salary of each category, estimated from the nationally representative Survey for Social Characterization in Chile [26].

c. Productivity losses: We used the human capital approach to estimate the productivity loss costs attributable to MSKD. We followed the described methodology in Virta et al. [27] that establishes that the income average appropriately reflects the value of economic goods produced by an individual. This assumption relies on individuals paid by their insurer representing their marginal productivity. Therefore, as the economic loss is associated with productivity, we estimated the product of the daily income of an individual multiplied by the number of sick leave days [28]. Data were obtained from the same health insurance database of sick leave, but, in this case, we did not exclude the first three sick leave days that are supposed to be paid by the employer.

It is important to highlight that although we used the same data to calculate sick leave and productivity losses, they are conceptually different. The first is a direct cost that health insurers must pay individuals a subsidy while they are sick. The second is an indirect cost borne by society due to the productivity loss.

The consequences presented in this analysis were the loss in health states utilities (LHSU) attributable to chronic pain, the prevalence of patients with chronic pain who suffer depression/anxiety, and the number of days of sick leave. LHSU was estimated through the modeling exercise. The reported disutility for the disability related to the pain of each of the five MSKD considered was multiplied by the chronic pain-attributable fraction (PAF), adding the sum across the time horizon (T). It represents the proportion of the disutility caused by chronic pain and the number of n individuals alive (n) at cycle t $[\sum_t^T disutilities_i \times PAF_i \times n_t]$. Disutility was obtained from the Zitko P. et al. [19] study. Depression/anxiety prevalence was estimated using the *depression/anxiety caused by a chronic pain-attributable fraction* (dPAF) [16] and then multiplying by the number of prevalent cases for each health problem. The ENS 2016–17 could not provide information to estimate the anxiety caused by a chronic pain-attributable fraction for each health problem but was possible as an all MSKD estimate. We assume that all MSKD estimate was equivalent to each health problem. Our time horizon was one year; hence, when all utilities lost were multiplied by one, utilities became QALYs. Further, they were presented for the total prevalent population in one year.

Finally, to characterize second-order uncertainty for costs related to pain management, depression, and anxiety, we adopted a Bayesian approach using classical techniques of probabilistic sensitivity analysis suggested for economic evaluations. We ran a 10,000 Montecarlo simulation based on which we estimated 95% credibility intervals. Because of the estimates for sick leave and productivity losses we obtained directly from the data, we provide 95% confidence intervals. The list of parameters used to inform the decision model and their corresponding a priori distributions assigned to them are shown in Table 1.

**Table 1. Model parameters and parametric distributions.**

| Item | Value | Standard Error | Distribution | Reference |
|---|---|---|---|---|
| **Prevalence** | | | | |
| Lower Back Pain | | | | |
| Mild | 0.61% | 0.31% | Beta | [19] |
| Moderate | 3.42% | 0.64% | Beta | [19] |
| Severe | 1.79% | 0.39% | Beta | [19] |
| Very Severe | 1.00% | 0.25% | Beta | [19] |
| Chronic Shoulder Pain | | | | [19] |
| Mild | 0.08% | 0.03% | Beta | [19] |
| Moderate/Severe | 2.89% | 0.43% | Beta | [19] |
| Osteoarthritis of the Hip | | | | [19] |
| Mild | 1.80% | 0.25% | Beta | [19] |
| Moderate | 0.10% | 0.05% | Beta | [19] |
| Severe | 0.17% | 0.08% | Beta | [19] |
| Osteoarthritis of the Knee | | | | [19] |
| Mild | 3.01% | 0.45% | Beta | [19] |
| Moderate | 0.32% | 0.11% | Beta | [19] |
| Severe | 0.22% | 0.09% | Beta | [19] |
| Fibromyalgia | | | | [19] |
| Moderate | 1.37% | 0.36% | Beta | [19] |
| Severe | 3.41% | 0.41% | Beta | [19] |
| All musculoskeletal diseases | | | | [19] |
| Mild | 0.98% | 0.50% | Beta | [19] |
| Moderate | 9.90% | 8.19% | Beta | [19] |
| Severe | 11.55% | 9.98% | Beta | [19] |
| **Transition Probabilities** | | | | |
| Mild-Moderate | 18.41% | N/A | Bootstrap | [16] |
| Mild-Severe | 10.70% | N/A | Bootstrap | [16] |
| Moderate-Severe | 27.38% | N/A | Bootstrap | [16] |
| Moderate-Mild | 55.55% | N/A | Bootstrap | [16] |
| Severe-Mild | 33.15% | N/A | Bootstrap | [16] |
| Severe-Moderate | 59.14% | N/A | Bootstrap | [16] |
| Probability of Death | 0.04% | 0,01% | Beta | [21] |
| **Costs (USD per Month)** | | | | |
| Anxiety | $14.55 | $14.26 | Gamma | Point Estimate |
| Depression | $26.20 | $25.68 | Gamma | Point Estimate |
| Lower Back Pain | | | | |
| Mild | $6.97 | $6.83 | Gamma | Point Estimate |
| Moderate | $13.75 | $13.48 | Gamma | Point Estimate |
| Severe | $54.22 | $53.13 | Gamma | Point Estimate |
| Chronic Shoulder Pain | | | | |
| Mild | $8.65 | $8.48 | Gamma | Point Estimate |
| Moderate | $15.56 | $15.25 | Gamma | Point Estimate |
| Severe | $51.76 | $50.72 | Gamma | Point Estimate |
| Osteoarthritis of the Hip | | | | |
| Mild | $8.60 | $8.43 | Gamma | Point Estimate |
| Moderate | $15.07 | $14.77 | Gamma | Point Estimate |
| Severe | $61.29 | $60.07 | Gamma | Point Estimate |

*(Continued)*

**Table 1.** (*Continued*)

| Item | Value | Standard Error | Distribution | Reference |
|---|---|---|---|---|
| Osteoarthritis of the Knee | | | | |
| Mild | $8.50 | $8.33 | Gamma | Point Estimate |
| Moderate | $15.07 | $14.77 | Gamma | Point Estimate |
| Severe | $61.29 | $60.07 | Gamma | Point Estimate |
| Fibromyalgia | | | | |
| Mild | $6.97 | $6.83 | Gamma | Point Estimate |
| Moderate | $9.70 | $9.51 | Gamma | Point Estimate |
| Severe | $83.26 | $81.60 | Gamma | Point Estimate |
| All musculoskeletal diseases | | | | |
| Mild | $7.94 | $7.78 | Gamma | Point Estimate |
| Moderate | $13.83 | $13.56 | Gamma | Point Estimate |
| Severe | $62.37 | $61.12 | Gamma | Point Estimate |
| **Productivity Loss Estimation** | | | | |
| Expected Costs (per sick leaves) | | | | |
| Lower Back Pain | $265.69 | $0.42 | Lognormal | Public Dataset 2019 |
| Chronic Shoulder Pain | $527.89 | $3.97 | Lognormal | Public Dataset 2019 |
| Osteoarthritis of the Hip | $793.94 | $5.43 | Lognormal | Public Dataset 2019 |
| Osteoarthritis of the Knee | $617.17 | $3.42 | Lognormal | Public Dataset 2019 |
| Fibromyalgia | $468.02 | $3.49 | Lognormal | Public Dataset 2019 |
| All musculoskeletal diseases | $312.42 | $0.50 | Lognormal | Public Dataset 2019 |
| **Absenteeism Costs Estimation** | | | | |
| Expected Costs (per sick leaves) | | | | |
| Lower Back Pain | $183.03 | $0.40 | Lognormal | Public Dataset 2019 |
| Chronic Shoulder Pain | $403.75 | $3.66 | Lognormal | Public Dataset 2019 |
| Osteoarthritis of the Hip | $725.79 | $5.62 | Lognormal | Public Dataset 2019 |
| Osteoarthritis of the Knee | $518.66 | $3.33 | Lognormal | Public Dataset 2019 |
| Fibromyalgia | $371.61 | $3.27 | Lognormal | Public Dataset 2019 |
| All musculoskeletal diseases | $225.84 | $0.48 | Lognormal | Public Dataset 2019 |
| **Expected Absenteeism Days (sick leaves)** | | | | |
| Lower Back Pain | 8.51 | 0.01 | Lognormal | Public Dataset 2019 |
| Chronic Shoulder Pain | 15.31 | 0.10 | Lognormal | Public Dataset 2019 |
| Osteoarthritis of the Hip | 27.12 | 0.15 | Lognormal | Public Dataset 2019 |
| Osteoarthritis of the Knee | 19.41 | 0.09 | Lognormal | Public Dataset 2019 |
| Fibromyalgia | 10.40 | 0.11 | Lognormal | Public Dataset 2019 |
| All musculoskeletal diseases | 9.94 | 0.01 | Lognormal | Public Dataset 2019 |
| **Consequences** | | | | |
| Lower Back Pain | | | | |
| Disability Related to Pain | | | | |
| Mild | 2.00% | 1.22% | Beta | [19] |
| Moderate | 5.40% | 2.24% | Beta | [19] |
| Severe | 32.20% | 11.40% | Beta | [19] |
| Chronic Shoulder Pain | | | | |
| Disability Related to Pain | | | | |
| Mild | 2.80% | 1.43% | Beta | [19] |
| Moderate/Severe | 11.70% | 4.23% | Beta | [19] |
| Osteoarthritis of the Hip | | | | |
| Disability Related to Pain | | | | |

(*Continued*)

**Table 1.** (Continued)

| Item | Value | Standard Error | Distribution | Reference |
|---|---|---|---|---|
| Mild | 2.30% | 1.22% | Beta | [19] |
| Moderate | 7.90% | 2.86% | Beta | [19] |
| Severe | 16.50% | 6.12% | Beta | [19] |
| Osteoarthritis of the Knee | | | | |
| Disability Related to Pain | | | | |
| Mild | 2.30% | 1.22% | Beta | [19] |
| Moderate | 7.90% | 2.86% | Beta | [19] |
| Severe | 16.50% | 6.12% | Beta | [19] |
| Fibromyalgia | | | | |
| Disability Related to Pain | | | | |
| Moderate | 31.70% | 11.43% | Beta | [19] |
| Severe | 51.80% | 17.14% | Beta | [19] |
| All musculoskeletal diseases | | | | |
| Disability Related to Pain | | | | |
| Mild | 2.00% | 1.22% | Beta | [19] |
| Moderate | 5.40% | 2.24% | Beta | [19] |
| Severe | 27.20% | 9.74% | Beta | [19] |
| Anxiety-Pain-attributable fraction | 5.30% | 1.35% | Beta | [29] |
| Lower Back Pain | 0.64% | 0.35% | Beta | [19] |
| Chronic Shoulder Pain | 1.29% | 0.90% | Beta | [19] |
| Osteoarthritis of the Hip | 0.48% | 0.26% | Beta | [19] |
| Osteoarthritis of the Knee | 0.54% | 0.35% | Beta | [19] |
| Fibromyalgia | 13.15% | 3.66% | Beta | [19] |
| All musculoskeletal diseases | 16.92% | 3.30% | Beta | [19] |

## Results

The total expected annual cost attributable to MSKD was USD 943,413,490 including costs of chronic pain management, depression, anxiety, and direct costs for the health system of sick leaves and indirect productivity losses. As mentioned before, they were restricted to the five diseases analyzed in this study (knee osteoarthritis (knee OA), hip osteoarthritis (hip OA); lower back pain (LBP); shoulder pain; and fibromyalgia (FM)). From the total, 80.9% corresponds to costs attributable to the five diseases studied. The remaining 19.1% corresponds to other musculoskeletal diseases not estimated in this study.

Regarding the five diseases analyzed in this study, the health problem with the highest total cost was low back pain (USD 320,952,132), followed by fibromyalgia (USD 143,710,082), Knee OA (USD 129,710,868), shoulder pain (USD 92,465,614), and Hip OA (USD 76,827,711) (Fig 3).

The cost for therapeutic management represents 72.6% of the total cost (USD 685,598,439), including chronic pain, depression, and anxiety. Chronic pain clinical care, in terms of direct costs, 82.5% of the total cost is attributable to the five diseases studied. Low back pain was the health problem with the highest cost among all other diseases (USD 187,908,323), representing 27.4% of the cost. In contrast, the lowest cost was from hip OA (USD 64,545,447), meaning 9.4% of the cost. Interestingly, among the five health problems studied, fibromyalgia is associated with the highest costs due to depression attributable to the condition, reaching more than USD 9.2 million per year approximately. Second, depression clinical care represents 7.51%, and third related anxiety clinical care represents 0.47% (USD 4,471,935) of the total cost of the

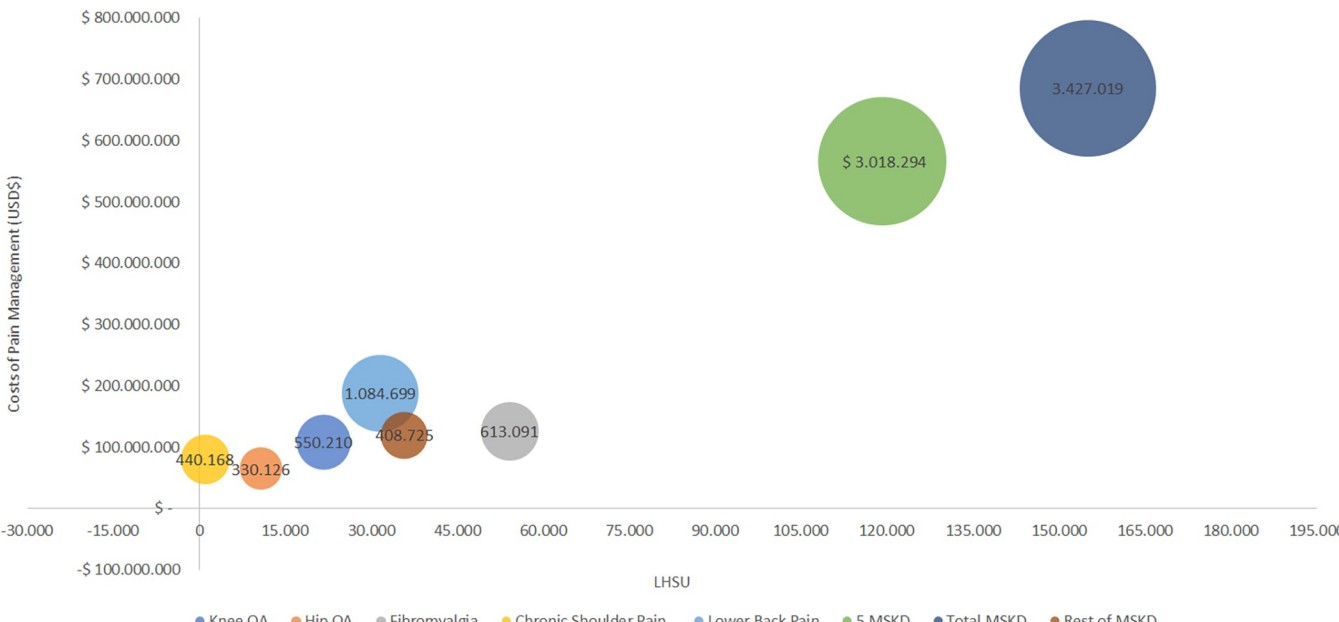

**Fig 3. Cost and consequences plot of chronic pain.** The figure represents the most significant costs (pain management) and consequences (LHSU) outcomes. The population of each health problem has been represented through the circle size.

therapeutic management. Interestingly, among the five health problems studied, low back pain is associated with the highest costs due to depression attributable to the condition, reaching more than USD 1.3 million per year approximately. These results with their corresponding 95% credibility intervals or 95% confidence intervals are presented in Table 2.

The estimates for sick leaves payments account for 8% of the total costs, been the highest cost was attributable to low back pain (USD 50,018,592). This magnitude was 5.7 times greater than the next more costly health problem, knee OA (USD 8,756,456). The cost of low back pain is reflected in the high number of sick leaves ─273,297─ and a total of 2,324,466 days paid sick leave days. Despite no differences between the frequency of sick leave licenses, the public health insurance takes 84% of this cost, which provides coverage to 78% of the Chilean population and reports the highest number of sick leaves issued (90% approximately). In contrast, private health insurance (ISAPREs) explains only a minor part of sick leaves payments. For both insurances, there is no distributive effect associated with a particular MSKD.

Regarding the cost from productivity losses, they represent 12% of the total cost. From the five suited diseases, low back pain has the more significant costs (USD 80,869,957), representing 72.9% of the total cost in productivity losses. Estimated costs from beneficiaries of the public health insurance were based on 3,147,509 days versus 254,994 days from private insurance beneficiaries. It is worth mentioning that productivity and sick leave payment costs include only five MSKD related to chronic pain, so the costs could be even higher.

As explained in the methods, expressed consequences through specific indicators. Thus, the consequences of pain were measured as losses in health state utilities for the affected population. The total expected loss was estimated at 155,036 QALYs prevalent, and the disease with the highest loss was fibromyalgia (54,172 QALYs prevalent), followed by low back pain (30,832 QALYs prevalent) (Fig 4). Another consequence of pain related to the specific MSKD is depression. Our estimates indicated that 2,689,958 episodes of depression and 303,997 cases of anxiety were attributed to pain. Finally, the total number of sick leaves days estimated from the official sources was 3,147,509.

**Table 2. Cost and consequences according to each health problem.**

| | | Knee OA | Hip OA | Fibromyalgia | Shoulder Pain | Lower Back Pain | Five MSKD | Rest of MSKD | Total MSKD |
|---|---|---|---|---|---|---|---|---|---|
| | | **Outcomes Expected Costs (Thousands USD$)** | | | | | | | |
| **Pain Management** | **Mean** | 108,00 | 64,55 | 125,70 | 80,03 | 187,91 | 566,17 | 119,43 | 685,60 |
| | **CrI 2.5%** | 19,41 | 11,88 | 21,01 | 14,28 | 36,05 | | | 139,49 |
| | **CrI 97.5%** | 295,30 | 177,71 | 374,17 | 217,26 | 500,06 | | | 1,755,631 |
| **Depression** | **Mean** | 329,00 | 192,00 | 9,27 | 648,00 | 812,00 | 11,26 | 59,60 | 70,86 |
| | **CrI 2.5%** | 5,00 | 4,00 | 232,00 | 7,00 | 15,00 | | | 2,06 |
| | **CrI 97.5%** | 1,60 | 883,00 | 37,36 | 3,22 | 3,69 | | | 265,39 |
| **Anxiety** | **Mean** | 671,00 | 459,00 | 732,00 | 548,00 | 1,34 | 3,75 | 718,00 | 4,47 |
| | **CrI 2.5%** | 18,00 | 9,00 | 0,00 | 13,00 | 35,00 | | | 13,00 |
| | **CrI 97.5%** | 2,65 | 1,81 | 2,84 | 2,25 | 5,40 | | | 17,29 |
| **Absenteeism** | **Mean** | 8,76 | 5,20 | 3,12 | 4,41 | 50,02 | 71,50 | N/D | N/D |
| | **CI 2.5%** | 8,61 | 5,10 | 3,05 | 4,30 | 49,73 | 71,09 | N/D | N/D |
| | **CI 97.5%** | 8,90 | 5,30 | 3,19 | 4,51 | 50,31 | 71,90 | N/D | N/D |
| **Productivity Losses** | **Mean** | 11,96 | 6,43 | 4,89 | 6,84 | 80,87 | 110,99 | N/D | N/D |
| | **CI 2.5%** | 11,81 | 6,34 | 4,80 | 6,73 | 80,54 | 110,53 | N/D | N/D |
| | **CI 97.5%** | 12,11 | 6,52 | 4,98 | 6,95 | 81,20 | 111,45 | N/D | N/D |
| | | **Outcomes Consequences** | | | | | | | |
| | | Knee OA | Hip OA | Fibromyalgia | Shoulder Pain | Lower Back Pain | Five MSKD | Rest of MSKD | Total MSKD |
| **LHSU** | **Mean** | 21,720 | 10,810 | 54,172 | 1,023 | 31,523 | 119,249 | 35,787 | 155,036 |
| | **CrI 2.5%** | 1,358 | 2,234 | 21,442 | 0 | 3,738 | - | - | 75,493 |
| | **CrI 97.5%** | 69,906 | 27,048 | 106,081 | 10,581 | 91,016 | - | - | 264,324 |
| **Depression** | **Mean** | 12,892 | 7,483 | 357,469 | 24,252 | 30,832 | 432,927 | 2,257,030 | 2,689,958 |
| | **CrI 2.5%** | 1,677 | 1,545 | 152,123 | 2,555 | 6,172 | - | - | 1,660,160 |
| | **CrI 97.5%** | 36,097 | 18,908 | 677,905 | 73,186 | 76,796 | - | - | 3,935,292 |
| **Anxiety** | **Mean** | 45,804 | 31,420 | 50,921 | 37,330 | 93,427 | 258,903 | 45,095 | 303,997 |
| | **CrI 2.5%** | 20,291 | 13,827 | 23,021 | 15,548 | 44,302 | - | - | 164,106 |
| | **CrI 97.5%** | 85,108 | 58,176 | 93,693 | 71,762 | 168,972 | - | - | 484,046 |
| **Absenteeism** | **Mean** | 387,961 | 221,281 | 182,945 | 205,647 | 2,677,424 | 3,147,509 | - | - |

Footnote: OA: Osteoarthritis; MSKD: musculoskeletal disease; CrI: Credible interval

## Discussion

The main findings of this study reflect the magnitude of the cost of chronic pain for the country. The estimated total cost was USD 943,413,490, with low back pain and fibromyalgia generating the highest cost to the system. Added to this, the therapeutic management of the five diseases included in this study: knee osteoarthritis (knee OA), hip osteoarthritis (hip OA); lower back pain (LBP); shoulder pain; and fibromyalgia (FM), represents 72.6% of the total cost (USD 685,598,439). This cost includes the clinical approach for depression and anxiety, two prevalent health problems in people who suffer from chronic pain. Finally, for QALYs, a more significant impact is observed in people with low back pain and fibromyalgia. As shown in our results, these two diseases have greater costs in chronic pain probably because they are long-standing health problems, with a high component of psychological needs and multiple barriers both from the health system and social for their approach.

Regarding costs, our study shows a strong correlation between costs of therapeutic management and the cost derived from labor sick leave and productivity losses (excluding FM). In the way therapeutic or clinical care costs increase, there is an increase in productivity losses and

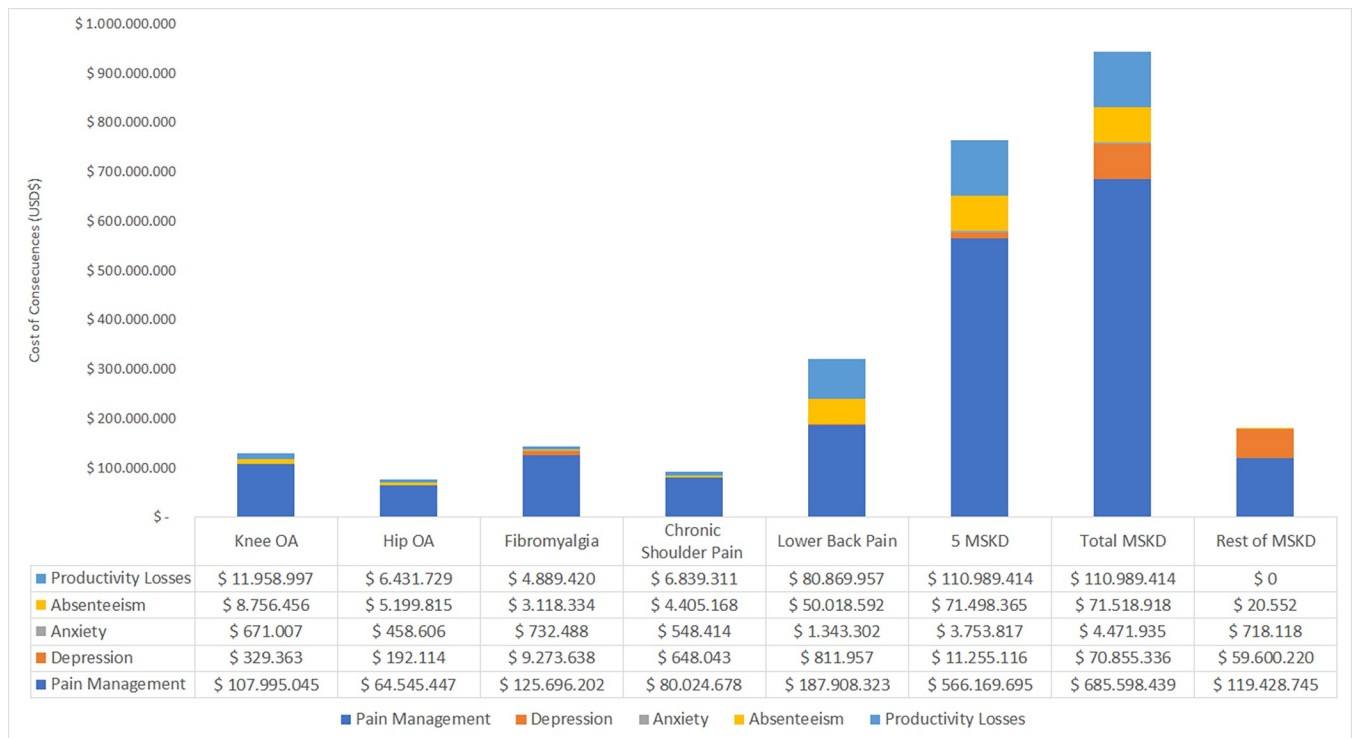

**Fig 4. Graphical comparison of every outcome for each health problem, further total MSKD, the sum of health problems, and an estimation of the rest of MSKD.**

labor sick leave. It can be explained by the long-term care these patients go through, together with the scarcity of access to a psychological and physical approach [30]. However, the impact in costs is not only for the health sector but also for others, where social security and other governmental institutions are affected by increased productivity losses and sick leave days.

In addition, it might be even underestimating the high costs found in this study. As shown in a study about hospital financial situation [31], calculated health services costs are significantly higher than the government fees, as they cover only 43% of the actual cost. It indeed produces an underestimation of the cost, affecting the estimations provided in this study. In other words, the costs estimated in this study are probably higher in the real context. Then, it is an urgent challenge for decision-makers and economists to adjust the costs associated with chronic pain in a lesser way since it includes a significant fiscal expense.

Regarding productivity losses, this study has the strength that used the human capital approach for cost estimation, which is the most used methodology across economic evaluation and cost-of-illness literature [32]. Although the human capital method has been criticized for overestimating costs, the other alternative method, the friction method, underestimates costs [33]. The first method values the productivity of individuals according to their income and, the loss of productivity is limited to the adaptation period called the friction period. In contrast, the second includes the employer perspective. As the literature reports that the human capital approach overestimates indirect costs in the long run (includes potential costs) [34, 35], we are considering a short analysis horizon (1 year).

About 28% of the Chilean labor force works without an established work contract; hence it is not registered in social security systems [36]. In this context, we could be underestimating the real magnitude of productivity losses attributable to MSKD because we are estimating

productivity losses from absenteeism days — which are proxied by sick leaves days from formal work contracts.

Another study's strength was accessing private and public health insurance databases that facilitated the estimations and comparisons between insurance productivity losses costs. Thus, costs estimated in our work represent 12% of the total cost, and they come mainly from the public health insurer, and even more, from those with lower income (FONASA A, B, and C). Differ from the magnitude reported by Vargas C et al. [16]. for the Chilean setting, where the productivity losses account only for 4% of the total costs. The difference can be explained because they are estimated only with the private health insurance database. Consequently, the authors assumed that the demand and prescription of medical licenses were equivalent between private and public insurance to estimate productivity costs.

On the other hand, the clinical care approach is not clearly described in the entire health network. For example, many patients experience refractory pharmacological treatment. The access and coverage for individualized treatment are basic, based mainly on NSAIDs through health services baskets for single musculoskeletal diseases regulated in quantity and time [36]. Therefore, treatment is interrupted and fragmented, failing to generate an adequate clinical response in people with chronic pain, increasing their consumption of resources in the healthcare network [12]. It is reflected in the costs analyzed in this study, where therapeutic management of chronic pain is the more significant proportion of the total costs. The Ministry of Health has recently published a technical reference for the chronic pain approach in primary care services [30], undoubtedly an important step. Though, it remains to be implemented in real practice and evaluate the impact on costs.

Future studies could complement this study from the perspective of the multimorbidity approach. Today, Chile and other countries are starting to respond and reorganize their care services to give a more effective approach [11, 37, 38]. Although its impact is well described in the literature [1, 39], further studies could estimate the cost of chronic pain as part of the cost of multimorbidity. Finally, the COVID-19 pandemic has further accentuated health problems widely, making this subject interesting to study in the future and generating recommendations for yet to come pandemics.

## Conclusion

Chronic pain is a health problem that has been increasingly important in our country and others in the region in the last few years. The high cost reflects the high prevalence, the fragmented approach, and the inadequate care delivery that the health system offers today for people with chronic pain. The importance of generating the necessary actions to provide better access, coverage, and financing for this health problem has been reflected. However, it must go along with a reorganization in the care delivery model to improve the basic chronic pain clinical approach. Driving through both actions could be today the best strategy for the health system to offer a cost-effective approach and address the challenges of this health problem more efficiently.

## Author Contributions

**Conceptualization:** Manuel A. Espinoza, Norberto Bilbeny, Tomas Abbott, Cesar Carcamo, Pedro Zitko, Paula Zamorano, Carlos Balmaceda.

**Data curation:** Manuel A. Espinoza, Tomas Abbott, Carlos Balmaceda.

**Formal analysis:** Manuel A. Espinoza, Tomas Abbott, Pedro Zitko, Carlos Balmaceda.

**Investigation:** Cesar Carcamo.

**Methodology:** Manuel A. Espinoza, Tomas Abbott, Carlos Balmaceda.

**Project administration:** Paula Zamorano.

**Resources:** Cesar Carcamo, Carlos Balmaceda.

**Supervision:** Paula Zamorano.

**Validation:** Manuel A. Espinoza, Norberto Bilbeny, Pedro Zitko, Paula Zamorano, Carlos Balmaceda.

**Visualization:** Norberto Bilbeny, Cesar Carcamo, Pedro Zitko.

**Writing – original draft:** Manuel A. Espinoza, Tomas Abbott, Paula Zamorano, Carlos Balmaceda.

**Writing – review & editing:** Norberto Bilbeny, Cesar Carcamo, Pedro Zitko.

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
