## [Decision Letter · Decision Letter 0]

4 Feb 2022

PONE-D-21-34991Cost analysis of chronic pain due to musculoskeletal disorders in ChilePLOS ONE

Dear Dr. Espinoza,

Thank you for submitting your manuscript to PLOS ONE. After careful consideration, we feel that it has merit but does not fully meet PLOS ONE’s publication criteria as it currently stands. Therefore, we invite you to submit a revised version of the manuscript that addresses the points raised during the review process.

We look forward to receiving your revised manuscript.

Kind regards,

Chengappa Kavadichanda

Academic Editor

PLOS ONE

Journal Requirements:

“no”

“no”

Reviewers' comments:

Reviewer's Responses to Questions

**Comments to the Author**

1. Is the manuscript technically sound, and do the data support the conclusions?

Reviewer #1: Yes

Reviewer #2: Partly

2. Has the statistical analysis been performed appropriately and rigorously? 

Reviewer #1: Yes

Reviewer #2: Yes

3. Have the authors made all data underlying the findings in their manuscript fully available?

Reviewer #1: Yes

Reviewer #2: No

4. Is the manuscript presented in an intelligible fashion and written in standard English?

Reviewer #1: Yes

Reviewer #2: Yes

5. Review Comments to the Author

Reviewer #1: The authors aim to estimate the direct and indirect economic burden of chronic pain in Chile focusing on 5 diseases with LBA and FMS contributing to the highest cost borne for chronic pain amongst MsKDs.

The manuscript needs revision in the following aspects

The entire manuscript is wordy and has language errors. Writing needs to be improved to add clarity to the manuscript.

The methodology is confusing. Type of study, subheadings, number of patients assessed etc need to be mentioned. A flow chart can be added for better clarity

The abbreviations in the tables need expansion. A common way of writing should be followed ($ mentioned in the upper part of table 2 only)

The discussion can be improved upon by comparing it with other similar studies.

Reviewer #2: The manuscript represents the expected cost, i.e., total cost, sick leave payment and productivity losses associated with chronic pain and related musculoskeletal diseases in the Chilean adult population. Apart from total therapeutic cost, the study has shown the impact of pain management with respect to social security and productivity losses concerning human capital. Following are the comments that could be considered for improving the manuscript:

Major comments:

1. The perspective of cost analysis could be explained more explicitly.

2. Justification shall be given in adopting a time horizon of 1 year as the musculoskeletal disorders are progressive in nature and last for a lifetime.

3. Although the model was adapted from the previous study, the present manuscript should include a schematic representation of the Markov model.

4. Does the model account for drug-induced side effects, particularly when NSAIDs, opioid analgesics, and oral corticosteroids are used to treat chronic refractory pain? Because failure could lead to underestimation of cost.

5. Whether are the costs or resources incurred for invasive management of patients with server chronic pain included while estimating the total therapeutic cost?

6. The level of utilization has a significant impact on unit cost estimates. Please explain how COVID-19 management affects the utilization of pain management services in Chile and how this was taken into account when evaluating unit cost for the current study.

7. Does the statement in lines 299–301 constitute limitation in evaluating the cost of sick leaves? Please clarify.

Minor comments:

8. Can the decimal points in table 1 be replaced with a dot instead of a comma?

9. The use of upper and lower case in table 1 is inconsistent.

10. Sentences in the following lines need to be rephrased or corrected: Lines 302–303; line 74.

6. PLOS authors have the option to publish the peer review history of their article (what does this mean?). If published, this will include your full peer review and any attached files.

Reviewer #1: No

Reviewer #2: No

---

## [Author Response · Author response to Decision Letter 0]

8 Jun 2022

Dear Editor, 

After making a careful review of our manuscript, based on the suggestions made by the reviewers, we have proceeded to send a new version of the manuscript “Cost analysis of chronic pain due to musculoskeletal disorders in Chile”. You will find in the system a new consolidated manuscript and the document with track changes. We thank the reviewers for all the comments that certainly helped us to enhance our manuscript. 

Please find in revision notes document the answers to each comment made by the reviewers and to the manuscript.

Kind Regards,

Dr. Manuel Espinoza, MD MSc PhD

On behalf of the authors

1 Associate Professor, Department of Public Health, Pontificia Universidad Católica de Chile, Diagonal Paraguay 362, Piso 2, Santiago, Chile.

Email: manuel.espinoza@uc.cl

---

## [Decision Letter · Decision Letter 1]

8 Aug 2022

PONE-D-21-34991R1Cost analysis of chronic pain due to musculoskeletal disorders in ChilePLOS ONE

Dear Dr. Espinoza,

Thank you for submitting your manuscript to PLOS ONE. After careful consideration, we feel that it has merit but does not fully meet PLOS ONE’s publication criteria as it currently stands. Therefore, we invite you to submit a revised version of the manuscript that addresses the points raised during the review process.

We look forward to receiving your revised manuscript.

Kind regards,

Chengappa Kavadichanda

Academic Editor

PLOS ONE

Journal Requirements:

Additional Editor Comments (if provided):

Dear Authors,

We are grateful for the changes you have made. There is however a minor change suggested by Reviewer 1.

Reviewers' comments:

Reviewer's Responses to Questions

**Comments to the Author**

1. If the authors have adequately addressed your comments raised in a previous round of review and you feel that this manuscript is now acceptable for publication, you may indicate that here to bypass the “Comments to the Author” section, enter your conflict of interest statement in the “Confidential to Editor” section, and submit your "Accept" recommendation.

Reviewer #1: All comments have been addressed

2. Is the manuscript technically sound, and do the data support the conclusions?

Reviewer #1: Yes

3. Has the statistical analysis been performed appropriately and rigorously? 

Reviewer #1: Yes

4. Have the authors made all data underlying the findings in their manuscript fully available?

Reviewer #1: Yes

5. Is the manuscript presented in an intelligible fashion and written in standard English?

Reviewer #1: Yes

6. Review Comments to the Author

Reviewer #1: Thank you for making the revisions. The manuscript looks much better now.

Kindly expand the abbreviations in Table 1.

7. PLOS authors have the option to publish the peer review history of their article (what does this mean?). If published, this will include your full peer review and any attached files.

Reviewer #1: No

---

## [Author Response · Author response to Decision Letter 1]

12 Aug 2022

Reviewer: Thank you for making the revisions. The manuscript looks much better now. Kindly expand the abbreviations in Table 1.

Response:Thanks for your comment, we have replaced the abbreviations for the full text on table 1.

---

## [Editor Report · Decision Letter 2]

15 Aug 2022

Cost analysis of chronic pain due to musculoskeletal disorders in Chile

PONE-D-21-34991R2

Dear Dr. Espinoza,

We’re pleased to inform you that your manuscript has been judged scientifically suitable for publication and will be formally accepted for publication once it meets all outstanding technical requirements.

Kind regards,

Chengappa Kavadichanda

Academic Editor

PLOS ONE
---

## [Editor Report · Acceptance letter]

29 Sep 2022

PONE-D-21-34991R2 

Cost analysis of chronic pain due to musculoskeletal disorders in Chile 

Dear Dr. Espinoza:

I'm pleased to inform you that your manuscript has been deemed suitable for publication in PLOS ONE. Congratulations! Your manuscript is now with our production department. 

Kind regards, 

on behalf of

Dr. Chengappa Kavadichanda 

Academic Editor

PLOS ONE